



# Simulated photochemical response to observational constraints on aerosol vertical distribution over North China

Xi Chen[1,2], Ke Li[1,2*], Ting Yang[3], Xipeng Jin[1,2], Lei Chen[1,2], Yang Yang[1,2], Shuman Zhao[4], Bo Hu[3], Bin Zhu[5], Zifa Wang[3], Hong Liao[1,2]

[1] Joint International Research Laboratory of Climate and Environment Change, Jiangsu Key Laboratory of Atmospheric Environment Monitoring and Pollution Control, Collaborative Innovation Center of Atmospheric Environment and Equipment Technology, Nanjing University of Information Science and Technology, Nanjing 210044, China
[2] School of Environmental Science and Engineering, Nanjing University of Information Science and Technology, Nanjing 210044, China
[3] State Key Laboratory of Atmospheric Boundary Layer Physics and Atmospheric Chemistry (LAPC), Institute of Atmospheric Physics, Chinese Academy of Sciences, Beijing 100029, China
[4] College of Chemistry and Chemical Engineering, Dezhou University, Dezhou, 253023, China
[5] Collaborative Innovation Center on Forecast and Evaluation of Meteorological Disasters, Key Laboratory for Aerosol-Cloud-Precipitation of China Meteorological Administration, Nanjing University of Information Science and
Technology, Nanjing 210044, China

*Correspondence to*: Ke Li (keli@nuist.edu.cn)

**Abstract.** The significance of aerosol-photolysis interaction (API) in atmospheric photochemistry has been emphasized by studies utilizing box models and chemical transport models. However, few studies have considered the actual aerosol vertical distribution when evaluating API effects due to the lack of observations and the uncertainties in model simulation.

Herein, we integrated lidar and radiosonde observations with the chemical transport model (GEOS-Chem) to quantify the response of photochemistry to observational constraints on aerosol vertical distribution across different seasons in North China. The underestimation of aerosol optical depth (AOD) in lower layers and the overestimation in upper layers in GEOS-Chem model were revised. In response, photolysis rates changed following AOD, showing 33.4%–73.8% increases at the surface. Surface ozone increased by an average of 0.9 ppb and 0.5 ppb in winter and summer and the default API impact

on ozone reduced by 36%–56%. The weaker response in summer can be related to the compensatory effects of stronger turbulence mixing in the boundary layer. The long-lasting underestimation of ozone levels in winter was also greatly improved. Due to the enhanced photochemistry, $PM_{2.5}$ increased by 0.8 µg m$^{-3}$ in winter and 0.2 µg m$^{-3}$ in summer and increased strongly during pollution events with a maximum daily change of 16.5 µg m$^{-3}$ at Beijing station in winter. The weakened API effect in turn enhanced nitric acid formation by increasing atmospheric oxidizing capacity (13.7% increase

for OH radical) in high $NO_x$ emission areas and this helps explain the strong response of $PM_{2.5}$ in winter.



## 1 Introduction

China witnessed a significant reduction in fine particulate matter ($PM_{2.5}$) concentration over the past decade (Zhang et al.,
2019; Xiao et al., 2022), while ozone levels increased (Lu et al., 2018; Liu et al., 2023). Ozone pollution has emerged as a
critical air quality issue in China (Wang et al., 2022), especially in North China (Li et al., 2020), and is now spreading into
the colder seasons (Li et al., 2021a). Many studies concluded that the rapid decline in aerosol concentrations is a key factor
contributing to the increasing ozone trend ( Li et al., 2019b; Liu and Wang, 2020; Wang et al., 2020b; Li et al., 2021b; Ma
et al., 2021; Shao et al., 2021; Liu et al., 2023), highlighting the urgent need for further investigation on the interactions
between aerosols and ozone.

Surface ozone is produced through the photochemical reactions of volatile organic compounds (VOCs) and nitrogen oxides
($NO_x$), in which aerosol can also play an important role. Ozone chemistry can be affected via the heterogeneous uptake of
reactive gases on aerosol surfaces (Li et al., 2017; Li et al., 2018a; Li et al., 2019b; Ivatt et al., 2022). Furthermore, aerosols
can absorb or scatter solar radiation, thereby influencing ozone concentrations by both altering meteorological conditions
through aerosol–radiation feedback (ARF) (Xing et al., 2017; Qu et al., 2020; Zhao et al., 2023) and directly changing the
photolysis rates that is referred to as the aerosol–photolysis interaction (API). Overall, whether during multi-pollutant air
pollution episodes (Yang et al., 2022) or under seasonal mean conditions (Li et al., 2024), the impact of API on surface
ozone can overwhelm that of ARF. As such, we will focus on the API process in this study.

Early studies have confirmed the significance of API, employing either observation-based methods or chemical transport
models. Dickerson et al. (1997) reported that scattering aerosols can enhance ultraviolet radiation and accelerate
photochemical reactions while absorbing aerosols have the opposite effect. Subsequent studies further investigated the
impact of API during different types of pollution around the world, such as biomass burning (Mok et al., 2016; Baylon et
al., 2018; Li et al., 2018b), wildfire (Jiang et al., 2012), dust events (Kushta et al., 2014; Li et al., 2017), and anthropogenic
pollution (Li et al., 2011; Li et al., 2018a; Gao et al., 2020; Wu et al., 2020; An et al., 2021; Yang et al., 2022). In the highly
polluted areas of China, API can lead to 5%–12% decrease in surface ozone during anthropogenic pollution episodes in
warm seasons (Li et al., 2011; Yang et al., 2022) and 5%–20% decrease in cold seasons (Li et al., 2018a; Gao et al., 2020;
Wu et al., 2020). The effects of long-term aerosol changes on photolysis have also been estimated in China. Using the
troposphere ultraviolet and visible radiation (TUV) model, Zhao et al. (2021) found that the photolysis rate of nitrogen
dioxide ($J[NO_2]$) in North China increased by $1.3 \times 10^{-4}$ $s^{-1}$ per year from 2013 to 2019 due to the aerosol declines.
Consequently, the ozone concentration in North China increased by approximately 1 ppb during the period of rapid decline
in aerosol concentration (2013–2017) due to API, as indicated by WRF-CMAQ simulations (Liu and Wang, 2020). This
impact of API on long-term ozone trends was also confirmed by using the box model approach (Wang et al., 2020b; Ma et
al., 2021).



What's more, API can alter the atmospheric oxidizing capability and suppress the secondary aerosol formation, resulting
in changes in $PM_{2.5}$ concentrations. For example, API reduced $PM_{2.5}$ concentrations by 2.5% to 10% in North China Plain
during wintertime pollution events (Wu et al., 2020) and by 2.4% in Delhi during crop residue burning period (Chutia et
al., 2024). Li et al. (2024) reported that the impact of API could decrease $PM_{2.5}$ by 5.4 µg m$^{-3}$ in winter but increase $PM_{2.5}$
by 2.5 µg m$^{-3}$ in summer over eastern China. These results indicate that the effect of API on air quality changes should not
be overlooked both in short episodes and in the long-term period.

The key challenge in quantifying the impact of API on photochemistry is to reasonably represent the vertical distribution
of aerosols since their effects vary across different altitudes (Jacobson, 1998; Liao et al., 1999). Shi et al. (2021) input
observed $PM_{2.5}$ and black carbon profiles into an optical model and a one-dimensional radiative transfer model,
demonstrating that scattering aerosols can suppress photochemical reactions in the lower boundary layer but enhance them
in the upper boundary layer. However, other observation-based studies using radiative transfer models and box models
often rely on empirical vertical distributions of aerosols, which typically follow an exponential profile. Pietruczuk et al.
(2022) compared four different aerosol vertical distribution schemes and concluded that the choice of different scheme and
variations of aerosol column depth (AOD) have a comparable impact on photolysis rates. More importantly, the actual
vertical distribution of aerosols is often highly heterogeneous and far more complex than an exponential distribution (Hu
et al., 2020; Sun et al., 2023).

Previous modeling-based studies on the impact of API primarily compared the simulated surface $PM_{2.5}$ concentrations and
AOD to observations without validating the aerosol vertical profiles. As such, using simulated profiles can introduce great
uncertainty in the estimation of API effect as chemical transport models still struggle to reproduce the vertical distribution
of aerosols accurately (Kim et al., 2015b; Huszar et al., 2020; Zhai et al., 2021; Lu et al., 2023). Our previous work revealed
that the Goddard Earth Observing System Chemical Transport Model (GEOS-Chem) tends to underestimate the aerosol
extinction coefficient (AEC) by approximately 30% in the lower atmosphere during both summer and winter, while
overestimating it by 30% to 60% at higher altitudes in summer (Chen et al., 2024). Chen et al. (2022) verified the simulated
black carbon vertical profile from WRF-Chem and found an overestimation of 87.4% at the surface level, as well as an
underestimation of 14.9% between 300 m and 900 m. Similar bias in $PM_{2.5}$ profile simulation has also been reported by
Liu et al. (2021a). Introducing the urban canopy model (Kim et al., 2015b) or increasing minimum turbulent eddy
diffusivity (Liu et al., 2021a) can improve the simulation of surface aerosol concentration, but the overestimation in the
upper layer remained. Overall, it is urgent to better understand the API effects by considering the actual aerosol vertical
distributions.

In this study, we used continuous aerosol vertical distributions observed by ground-based lidar and radiosonde, along with
the GEOS-Chem model, to investigate the response of photochemistry to observational constraints aerosol vertical



distributions. Based on these observations, we modified the vertical distribution of aerosols in the photochemical module of GEOS-Chem over North China. The responses of photolysis rates, ozone, and $PM_{2.5}$ to observational constraints on aerosol vertical distribution through API across different seasons were quantitatively assessed by comparing control and sensitivity simulations. A description of the observations and model settings is presented in Section 2. Section 3 outlines the existing biases in the model and provides a detailed account of how the aerosol vertical distribution was modified.

Section 4 discusses the responses of photolysis, ozone, and $PM_{2.5}$ to this modification and their underlying mechanisms, followed by conclusions in Section 5.

## 2 Observations and model

### 2.1 Observations

To constrain the aerosol vertical distribution in the model, we utilized consecutive observations of aerosol extinction

coefficient (AEC) from a ground-based Mie-scattering lidar located at the Institute of Atmospheric Physics, Chinese Academy of Sciences (39.982°N, 116.385°E) in Beijing (Fig. 1a). The lidar scans every 15 min, and we employed quality-controlled AEC data at the wavelength of 532 nm with a vertical resolution of 30 m. Further details about the lidar system and data processing can be found in Wang et al. (2020a). Data from January and July 2017 were used to represent winter and summer seasons, respectively.

To verify the results constrained by lidar, we also utilized vertical $PM_{2.5}$ concentration profiles from radiosonde observations conducted in Baoding (Fig. 1a). The radiosonde data were collected every 3 hours from 17 June 2019 to 7 July 2019, at Baoding station (38.783°N, 115.5°E) with an altitude of 19 m. This radiosonde measurement provides high-resolution boundary layer profiles containing $PM_{2.5}$ and $PM_{10}$ concentrations, temperature, relative humidity (RH), and wind. A detailed description of the design of the radiosonde observations and the instruments used can be found in Li et al.

(2023). Surface observations of hourly ozone and $PM_{2.5}$ concentrations from the Ministry of Ecology and Environment (MEE) network were accessed at http://quotsoft.net/air (Wang, 2023) for the model evaluation.

### 2.2 GEOS-Chem simulation

The GEOS-Chem model version 13.3.3 was used in this study, employing a nested-grid simulation with a horizontal resolution of 0.5° × 0.625° over the region of 15–55°N and 70–140°E. Vertically, we selected the scheme with 47 layers,

where the first 8 layers are situated below 1 km, and the first 14 layers are below 2 km. MERRA-2 meteorological data from NASA and chemical boundary conditions from a global simulation with a resolution of 2° × 2.5° were used to drive the nested simulations. The boundary layer mixing, dry deposition, wet deposition schemes, and emissions were consistent with those used in our previous work (Chen et al., 2024). In GEOS-Chem, the photolysis rates are calculated by the updated



Fast-JX 7.0 scheme (Eastham et al., 2014), which accounts for the presence of aerosols, clouds, and absorbing gases. The

impacts of aerosols, including sulfate–nitrate–ammonium (SNA), organic aerosol (OA), black carbon (BC), sea salt, and

dust, on photolysis rates, are determined by aerosol optical properties, which are calculated using Mie code based on the

simulated three-dimensional aerosol concentrations. Here we modified the aerosol vertical distribution with observational

constraints after the calculation of AOD, and more details are presented in Section 3.

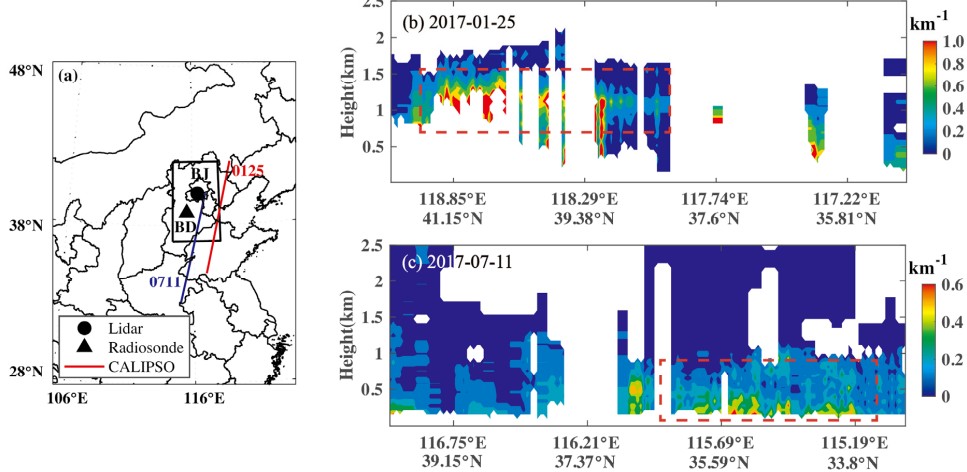

**Figure 1.** (a) Spatial distribution of lidar and radiosonde stations. BJ: Beijing; BD: Baoding. The black rectangle shows

the region where vertical distribution modification was performed. Vertical distribution of aerosol extinction coefficient

(532 nm) from CALIPSO on (b) 25 January 2017 and (c) 11 July 2017. The corresponding CALIPSO overpassing tracks

are marked with red and blue lines in (a).

The simulation periods included January and July 2017 for the lidar observation timeframe, and June–July 2019 for the

radiosonde observation timeframe. For each period, we performed three sensitivity experiments to examine the impact of

revised aerosol vertical distribution on photochemistry, as detailed below:

(1) BASE: The baseline simulation, which includes the API with the original aerosol vertical distribution in the model;

(2) REGION: Same as the BASE simulation, but incorporating observational constraints on aerosol vertical distribution

over North China (37–42°N, 114.375–118.125°E), wherein only the aerosol vertical distribution below 3 km was modified

in the calculation of photochemical rates based on lidar and radiosonde observations;

(3) API_OFF: Same as the BASE simulation, but with the effect of API turned off in the photochemical module.



By comparing the BASE and REGION simulations, we can quantify the photochemical response to observational constraints on aerosol vertical distribution over North China. The role of all aerosols in photolysis can be estimated by examining the differences between BASE and API_OFF simulations.

## 3 Revisiting model aerosol vertical distribution based on observational constraints

### 3.1 Existing biases in model aerosol vertical distribution

The model performance in simulating surface air pollutant concentrations and aerosol vertical distributions was evaluated. Table S1 presents the correlation coefficient and normalized mean bias (NMB) of ozone and $PM_{2.5}$ concentrations in North China. These statistics were calculated as averages across all observation stations in the region (37–42°N, 114.375–118.125°E) where modification of aerosol vertical distribution was applied. The GEOS-Chem generally reproduced the variation in surface concentrations in winter, achieving correlation coefficients of 0.69 for ozone and $PM_{2.5}$, with underestimations of 24.8% and 4.3% for ozone and $PM_{2.5}$, respectively. The model exhibits higher correlation coefficients (0.80–0.84) with the observations during summer, except for $PM_{2.5}$ in summer 2017. This seasonality of model performance aligns with validations reported in many previous studies (Kim et al., 2015a; Akimoto et al., 2019; Li et al., 2019a; Huszar et al., 2020; Zhai et al., 2021). The model simulated ozone best in the summer of 2019, with an NMB of −0.9%, but there was an overestimation of $PM_{2.5}$. Since we focused on the differences between sensitivity experiments, the model shows an acceptable performance in simulating surface air pollutants.

Figure 2 compares the average AOD profiles derived from the lidar observation with the BASE simulation. In winter, the AOD was underestimated by 20.8% and 30.1% in model layers 1 and 2, respectively, but it was subsequently overestimated until layer 10 (~1.29 km), with a maximum NMB of 18.9%. In summer, similarly, the AOD was underestimated by 3.4%–43.9% below layer 6 (~0.73 km) and was then overestimated by more than 50% above layer 9 (~1 km). As reported in our previous work, emission sources, optical properties, size distributions of aerosols, and RH are the key factors contributing to the model bias in AOD (Chen et al., 2024). We also compared the daily average $PM_{2.5}$ profiles from radiosonde data and GEOS-Chem (Fig. S1). Consistent with previous studies (Kim et al., 2015b; Liu et al., 2021a; Chen et al., 2022), the model has difficulty in accurately capturing the complex characteristics of aerosol vertical distribution. The comparisons of observed and simulated $PM_{2.5}$ profiles for three typical cases are presented in Fig. S1. Overestimation of $PM_{2.5}$ in the upper layer was common, while the model performance varied in the lower layer. The above simulation biases in aerosol vertical distribution underline the importance of observational constraints that will be elaborated in the following.





### 3.2 Modification of model aerosol vertical distribution

The ground-based lidar provided high-resolution AEC profiles, and we used the hourly average AEC profiles to conduct the modification after interpolating to model layers. The first model layer (approximately 58 m) is in the fade area of the lidar; therefore, we assumed the AEC in layer 1 to be the same as that in layer 2. We tested the impact of this assumption on ozone with the radiosonde data, which has observed data in layer 1 (Fig. S2). The results indicate that when this assumption was adopted, the effect of revised aerosol vertical distribution on ozone barely changed in both spatial

distribution and regional averages. Then, AOD for each layer can be obtained by multiplying AEC by the layer height. Subsequently, the observational AOD vertical distributions were incorporated into the model with the following equations:

$$r_i = \frac{AOD_{i,obs}}{AOD_{1-17,obs}}, \tag{1}$$

$$AOD_{i,new} = r_i \times AOD_{1-17,sim}, \tag{2}$$

where $r_i$ represents the ratio of AOD in layer $i$ to the total AOD within 3 km (layers 1 to 17 in the model), $AOD_{i,obs}$ is the

AOD in layer $i$ from observations, $AOD_{1-17,sim}$ refers to the column AOD below 3 km from the BASE simulation, and $AOD_{i,new}$ is the AOD in layer $i$ after modification. The AOD in GEOS-Chem model was summed up by sulfate, nitrate, ammonium, OA, BC, sea salt, and dust AOD, and the contribution of each aerosol component remained unchanged in the REGION simulation. This modification of aerosol vertical distribution was applied only within the photochemistry module when calculating photolysis rates.

Unlike lidar, radiosonde observations have no fade area near the surface. Therefore, we used $PM_{2.5}$ concentration profiles from radiosonde data collected in summer 2019 to validate the reliability of results derived from observational constraints based on lidar. Before constraining the model, we converted the $PM_{2.5}$ concentration profiles into AEC profiles using the Interagency Monitoring of Protected Visual Environments  (IMPROVE) algorithm (Pitchford et al., 2007). This algorithm estimates AEC based on measured species concentrations and has been widely validated in China (Cao et al., 2012; Tao et

al., 2014; Xiao et al., 2014; Bai et al., 2020). The $PM_{2.5}$ and $PM_{10}$ concentrations and RH profiles from radiosonde data, and the simulated proportions of aerosol components were utilized in the IMPROVE algorithm. Afterward, the model's aerosol vertical distribution was revised according to Eqs. (1–2), and it was updated every 3 hours.

The modification of aerosol vertical distribution was carried out over North China (37–42°N, 114.375–118.125°E) as shown in Fig. 1a. Based on the observed surface pollutant concentrations, we found a high correlation coefficient between

vertical observation stations and regional averages (Table S2). The correlation coefficients of ozone (0.88–0.95) and $PM_{2.5}$ (0.53–0.86) indicate that air pollution in North China usually occurs regionally. Moreover, we also used observations of AEC profiles from CALIPSO (Level 2, V4.20) to demonstrate that aerosol vertical distributions in North China exhibit





regional characteristics. Vertically, aerosols can occasionally be transported in the upper layer, as evidenced by an aerosol layer at approximately 1 km on January 25, 2017, extending for about 200 km across North China (Fig. 1b). Near the

surface, as observed on July 11, 2017 (Fig. 1c), high values in the lower layers were also observed over a regional extent of several hundred kilometers. Besides, we analysed the correlation of simulated aerosol vertical distributions at each grid point within the region and the observed grid point, finding that over half of the grid points exhibited significant correlation (Fig. S3). Li et al. (2022) also highlighted the regionality of the upper aerosol layer through joint observations from multiple radiosonde stations. As such, we believe that the aerosol vertical distribution in the modified region closely aligns with that

observed at the Beijing and Baoding stations, and the impact of aerosol vertical distribution modification on photochemistry in North China in this work may represent an upper estimate due to the regionally assumed profiles.

## 4 Impact of revised aerosol vertical distribution

Modifying aerosol vertical distribution can alter photolysis rates and thereby affect photochemical reactions. We evaluated the impact of revised aerosol vertical distribution on photolysis rates (Section 4.1), ozone (Section 4.2), and PM$_{2.5}$ (Section

4.3) across different seasons and discussed the influencing factors. It is noted that all impacts in this section were evaluated only during the daytime when photochemical reactions occur.

### 4.1 Impact of revised aerosol vertical distribution on photolysis rates

The photolysis of nitrogen dioxide (NO$_2$) and ozone are two major photolysis reactions in ozone production: $NO_2 + hv \rightarrow NO + O^3P$ ($\lambda < 420\ nm$) and $O_3 + hv \rightarrow O_2 + O^1D$ ($\lambda < 340\ nm$). Figure S4 shows the responses of their photolysis

rates ($J[NO_2]$ and $J[O^1D]$) at the surface layer to the observational constraints over North China. The photolysis rates increased only in the modified region, with a more pronounced increase from north to south. In winter, the regional average $\Delta J[NO_2]$ was 63.4% and $\Delta J[O^1D]$ was higher at 73.8%. The relative changes of $J[NO_2]$ and $J[O^1D]$ in North China were 33.4% and 41.8% in summer 2017, respectively, and spatial distributions were similar to those in winter. Compared with the results from lidar constraints, similar changes in photolysis rates were also found in radiosonde constraints. The $J[NO_2]$

and $J[O^1D]$ increased by 34.8% and 44.7% in summer 2019, respectively, with the difference in change rates compared to winter primarily attributed to the southern part of North China. In comparison, the impact of aerosol vertical distribution on photolysis rates in both winter and summer was much greater than the impact of absorbing OA (13.5%–18.1%) and BC (11.4%–12.5%) over North China as reported by Li and Li (2023). This further confirmed the importance of accurate representation of aerosol vertical distribution in evaluating API impact.





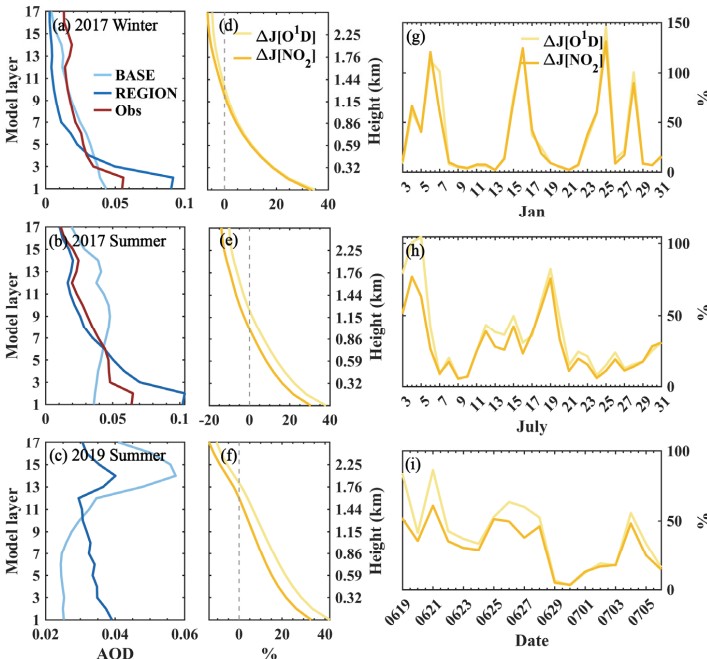

**Figure 2**. Vertical profiles of average (a–c) AOD, (d–f) relative changes of $J[NO_2]$, and $J[O^1D]$ in model layers at observation stations. The red, light, and dark blue lines represent observed, BASE simulated, and REGION simulated AOD, respectively. (g–i) Time series of daily average relative changes of surface $J[NO_2]$, and $J[O^1D]$ at observation stations. From top to bottom are for winter 2017 (a, d, g), summer 2017 (b, e, h), and summer 2019 (c, f, i), respectively. The averages are calculated using data during the daytime when photolysis occurs.

Figure 2 presents the vertical distributions of $\Delta J[NO_2]$ and $\Delta J[O^1D]$ at observation stations. With the observational constraints employed, the AOD at the lidar station increased below layer 4 in winter and decreased above this layer. Consequently, the photolysis rates increased in the lower layers and decreased in the higher layers, with the transition height approximately 0.5 km higher than AOD. Similarly, the modified AOD increased below layer 6 and layer 10 for the summer of 2017 and 2019 (Fig. 2b–c), which was more aligned with the observed exponential attenuation, and the changes in photolysis rates were consistent with the variations in AOD profiles. Differently, there was roughly a 5% greater increase in $J[O^1D]$ compared to $J[NO_2]$ in all layers during summer. In addition, the accuracy of column AOD simulations plays an important role in the impact of revised aerosol vertical distribution. Figure S5 shows the vertical distribution of AOD,



$\Delta J[NO_2]$ and $\Delta J[O^1D]$ under different scenarios of AOD simulation bias. When the column AOD was underestimated, the observationally constrained layer AOD was still lower than the observations. The revised vertical distribution was more consistent with observations when the column AOD was overestimated, but the revised AOD in lower layers was much higher than the observations. Consequently, the relative change of photolysis rates was approximately 3 times higher with overestimated AOD compared to underestimated AOD in winter, and 1.5 times higher in summer. In this work, we focused on aerosol vertical distribution and the improvement of AOD simulation should be promoted in future study.

We also compared the temporal changes in photolysis rates (Fig. 2g). The photolysis rates increased primarily during several pollution periods in winter, with peak values exceeding 100%. However, during clean days, aerosol vertical distribution had a limited impact on photolysis. The peak of the daily average photolysis rate changes in summer was notably lower than that in winter (Fig. 2h–i), yet the photolysis enhancement during summer was more consistent throughout the period.

To further verify the impacts of aerosol vertical distribution modification, we also compared the simulated surface $J[NO_2]$ to data at Xianghe station (39.76°N, 116.96°E) located in North China from Zhao et al. (2021) (Fig. 3). This dataset was reconstructed by observational $J[NO_2]$, total ultraviolet radiation, and troposphere ultraviolet and visible (TUV) radiation model, with an improved $R^2$ value of 0.94 between the observed and reconstructed data. The seasonal average $J[NO_2]$ in summer was higher than in winter, and it was greater in summer 2019 than in summer 2017 due to the decline in aerosol concentrations. More importantly, the BASE simulation underestimated observed-based $J[NO_2]$ during all simulation periods. In winter, the underestimation was 20.1%, but after the revision of aerosol vertical distribution, it turned into a 4.8% overestimation. A similar improvement in the underestimation of photolysis rates was also demonstrated in the summers of 2017 and 2019. These results indicate that revising aerosol vertical distribution can better simulate photochemical processes.

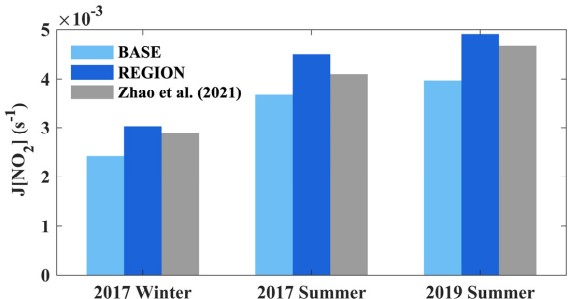

**Figure 3.** Simulated and observed average surface $\boldsymbol{J[NO_2]}$ at Xianghe station for winter 2017, summer 2017, and summer 2019. The seasonal average $\boldsymbol{J[NO_2]}$ from Zhao et al. (2021) is presented by grey bars.





## 4.2 Impact of revised aerosol vertical distribution on ozone

### 4.2.1 Wintertime

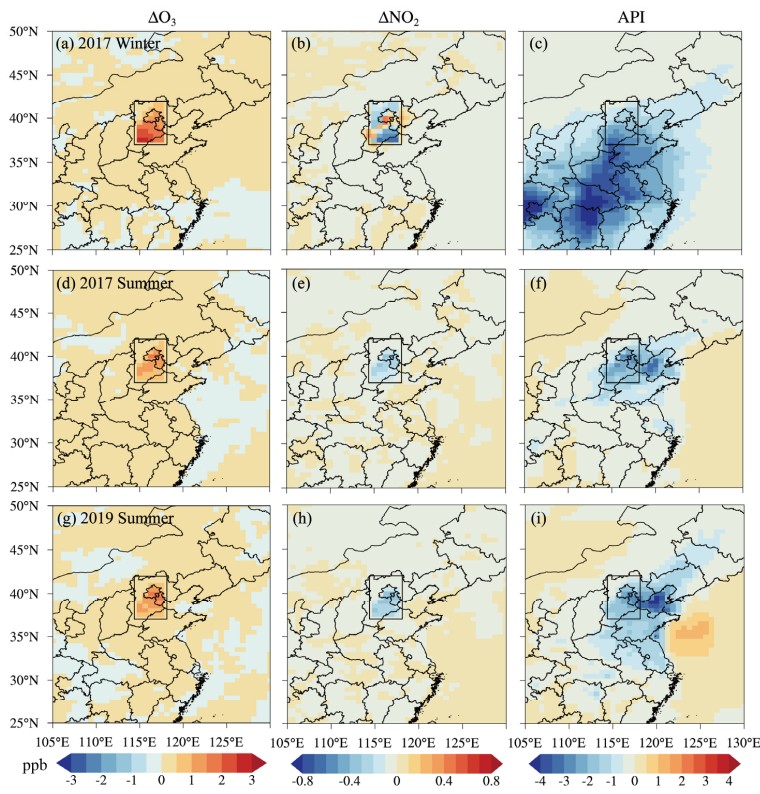


**Figure 4.** Simulated responses of daytime ozone and nitrogen dioxide (NO$_2$) to observational constraints on aerosol vertical distribution over North China (left and middle columns), along with the effect of aerosol–photolysis interaction (API) on daytime ozone (right column). From top to bottom are for winter 2017 (a–c), summer 2017 (d–f), and summer 2019 (g–i), respectively. The black rectangle shows the region where vertical distribution modification was applied.




Figure 4 shows the spatial distributions of changes in surface ozone and $NO_2$ in response to observational constraints on aerosol vertical distribution. Generally, as the aerosol layer moved from the upper layers to the lower layers, more solar

radiation reached the surface and promoted photolysis. Increased photolysis of $NO_2$ accelerated ozone production, leading to a rise in surface ozone levels and a corresponding decrease in $NO_2$. The variations in ozone displayed patterns that closely resembled the spatial distribution of changes in photolysis rates. In winter, ozone increased by an average of 0.9 ppb, with a maximum increase of 2.7 ppb in the modified region (Table 1). Figure 4c illustrates the default effect of API on ozone in winter, with a regional average of −1.6 ppb. The impact of revised aerosol vertical distribution on ozone can

reduce the API-induced ozone by 56%, implying that the vertical distribution is as important as the column AOD intensity in determining API. Moreover, the underestimation of ozone in North China was reduced from -25% in the BASE simulation to -17% in the REGION simulation (Table S1). This suggests that improving the accuracy of aerosol vertical distribution representativeness could mitigate the underestimation of ozone levels in winter.

**Table 1.** The regional maximum and average changes in ozone and $PM_{2.5}$ over North China for seasonal and daily mean. $O_3$_API is the default effect of aerosol–photolysis interaction (API) on ozone. Relative changes are also given in bracket.

| | Variable | Season | Maximum | Average |
|---|---|---|---|---|
| **Seasonal Mean** | $O_3$ (ppb) | 2017 winter | 2.7 | 0.9 |
| | | 2017 summer | 1.6 | 0.5 |
| | | 2019 summer | 2.3 | 0.7 |
| | $O_3$_API (ppb) | 2017 winter | −3.6 | −1.6 |
| | | 2017 summer | −2.8 | −1.4 |
| | | 2019 summer | −3.1 | −1.3 |
| | $PM_{2.5}$ ($\mu g\ m^{-3}$) | 2017 winter | 3.1 (1.2%) | 0.8 (0.2%) |
| | | 2017 summer | 0.7 (0.8%) | 0.2 (0.3%) |
| | | 2019 summer | 0.6 (1.1%) | 0.2 (0.5%) |
| **Daily Mean** | $O_3$ (ppb) | 2017 winter | 0.2–10.5 | 0.1–3.4 |
| | | 2017 summer | 0.4–7.1 | 0–1.9 |
| | | 2019 summer | 0.5–6.4 | 0.1–1.3 |
| | $PM_{2.5}$ ($\mu g\ m^{-3}$) | 2017 winter | 0.1–21.5 (0.4–5.3%) | −0.1–2.9(−0.6–1.5%) |
| | | 2017 summer | 0.2–2.9 (0.5–2.8%) | 0–0.7 (−0.3–1.0%) |
| | | 2019 summer | 0.1–2.3 (0.2–3.1%) | 0–0.6 (0–1.2%) |

 

Figures 5a and 5d further present the changes in daily average NO$_2$ and ozone along the longitude profile of the lidar station in winter. In line with the change in photolysis rates, the changes in NO$_2$ and ozone were also periodic. It is evident that the increase of NO$_2$ shown in Fig. 4b was not contributed by a single event. The surplus of NO$_x$ in the winter atmosphere, combined with the increase in ozone concentrations due to enhanced photolysis, led to a greater oxidation of NO. Therefore, NO$_2$ increased in the region with high NO$_x$ emission when ozone production was accelerated. For daily average, API led to a reduction of ozone by 0.1–6.2 ppb in North China. For reference, Xing et al. (2017) reported that API reduced daily maxima 1 h ozone by 2–8 ppb in January 2013 over North China Plain. With revised aerosol vertical distribution, ozone was primarily increased during the periods of 3–8 January, 15–18 January, and 23–28 January when photolysis was enhanced. The regional average daily ozone was increased by 0.1–3.4 ppb, and the regional maximum daily ozone was increased by 0.2–10.5 ppb (Table 1). Our results indicate that the impact of revised aerosol vertical distribution can greatly weaken the default API effect.

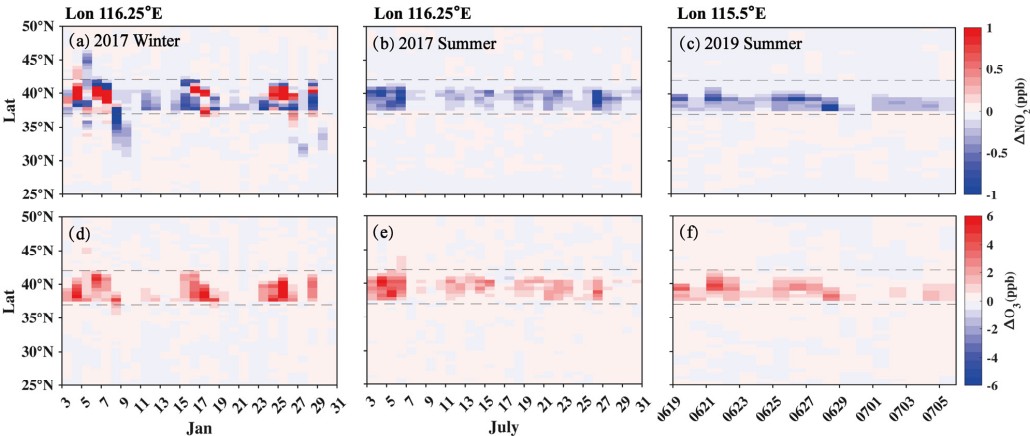

**Figure 5.** Simulated responses of daytime average ozone and nitrogen dioxide (NO$_2$) to observational constraints on aerosol vertical distribution along the longitude profiles of observation stations. From left to right are for winter 2017 (a, d), summer 2017 (b, e), and summer 2019 (c, f), respectively. The dotted lines indicate the latitude range of the modified region, and the longitudes of the stations are marked at the top.

### 4.2.2 Summertime

In summer, areas with increased photolysis rates experienced a decrease in NO$_2$ concentrations and an increase in ozone (Fig. 4d–e). However, the impact of revised aerosol vertical distribution on ozone concentrations was only half that in winter, with an average of 0.5 ppb and a maximum of 1.6 ppb in North China. The default effect of API on regional ozone



was −1.4 ppb, which was reduced by 36% after revising aerosol vertical distribution. Furthermore, ozone statistics in the REGION simulation show little difference from those in the BASE simulation (Table S1). These results indicate that the

significance of aerosol vertical distribution on ozone was less pronounced in summer than in winter. The response of ozone to observational constraints based on radiosonde data was similar to that based on lidar data in both spatial distribution and regional average. The regional average change in ozone during summer 2019 was 0.7 ppb, with a regional maximum of 2.3 ppb, both smaller than the changes in winter. The similar results constrained by these two sets of different observational data in summer confirm the reliability of this finding.

From the perspective of temporal variation, the increase in ozone and the decrease in $NO_2$ during summer corresponded to the periods of increased photolysis rates shown in Fig. 2. Ozone increased most significantly from 3 July to 6 July 2017, when photolysis rates increased most. The regional average daily ozone and the regional maximum daily ozone increased by 0–1.9 ppb and 0.4–7.1 ppb in summer 2017, 0.1–1.3 ppb and 0.5–6.4 ppb in summer 2019, respectively (Table 1). These results further confirmed that the impact of revised aerosol vertical distribution on ozone in summer is weaker compared

to winter.

### 4.2.3 Role of PBL mixing

As shown in Fig. 2, the modification of aerosol vertical distribution mainly increased the AOD in the lower layers while AOD was decreased in the upper layers across different seasons. With similar modifications of aerosol vertical distribution, the various responses of surface ozone concentration across different seasons are closely linked to planetary boundary layer

(PBL) mixing. Figure 6 and Figure 7 present the average diurnal variation in the vertical distribution responses for typical modification cases characterized by high planetary boundary layer height (PBLH) and low PBLH. The cases with high PBLH were in summer, while those with low PBLH were in winter (see detailed information in Table S3).

In Fig. 6a, the AOD decreased in the upper PBL and increased in the lower PBL after modification, with the PBLH reaching layer 15 (~2.2 km) at noon. Consequently, the photolysis rates changed following the changes in AOD, as shown in Fig.

6c and 6d. The changes in photolysis rates resulted in an increase in ozone in the lower PBL, a decrease in ozone production in the upper PBL, and a larger vertical gradient of ozone concentration. Then, the strong PBL mixing led to the high ozone concentrations at the surface being entrained into the upper PBL, partially offsetting the increase in surface ozone and, in some cases, even leading to a decrease in surface ozone. This kind of compensatory effect of PBL mixing on surface ozone in response to API was recently mentioned in previous studies (Gao et al., 2020; Shi et al., 2021; Yan et al., 2023). When

the PBLH started to decrease at 10:00 UTC, the photolysis rates increased in the PBL, and the response of surface ozone recovered. In Fig. 7, the reduction of AOD was primarily above the PBL, and the photolysis was promoted throughout the entire PBL, with the maximum PBLH reaching layer 6 (~0.73 km) at noon. As such, ozone increased at all layers in PBL. After sunset (18:00 UTC), the PBLH suddenly dropped to the lowest layer, and the increased ozone in the residual layer





reacted with the surplus NO in the atmosphere, increasing nighttime NO$_2$. Overall, the impact of revised aerosol vertical

distribution on surface ozone was larger under conditions with low PBLH. Due to stronger solar radiation and warming, PBL usually develops higher in summer than in winter. Hence, the weaker impact of revised aerosol vertical distribution on surface ozone in summer could be attributed to the compensatory effect of PBL mixing.

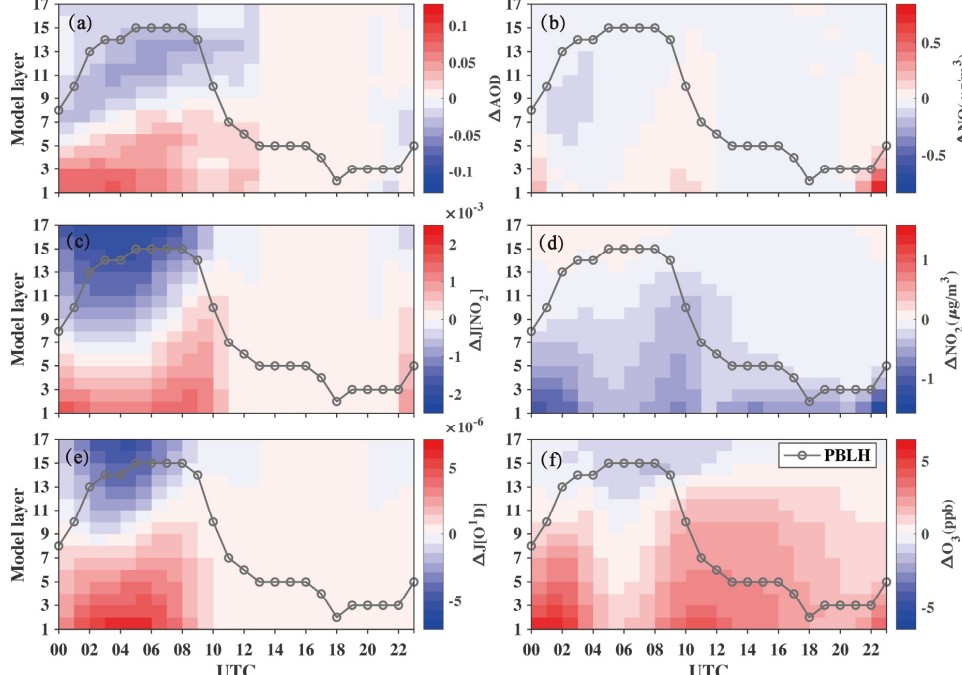

**Figure 6.** Diurnal variation in the vertical distribution responses of (a) AOD, (c) $J[NO_2]$, (e) $J[O^1D]$, (b) nitric oxide (NO),

(d) nitrogen dioxide (NO$_2$), and (f) ozone at the observation stations under conditions with high planetary boundary layer height (PBLH).



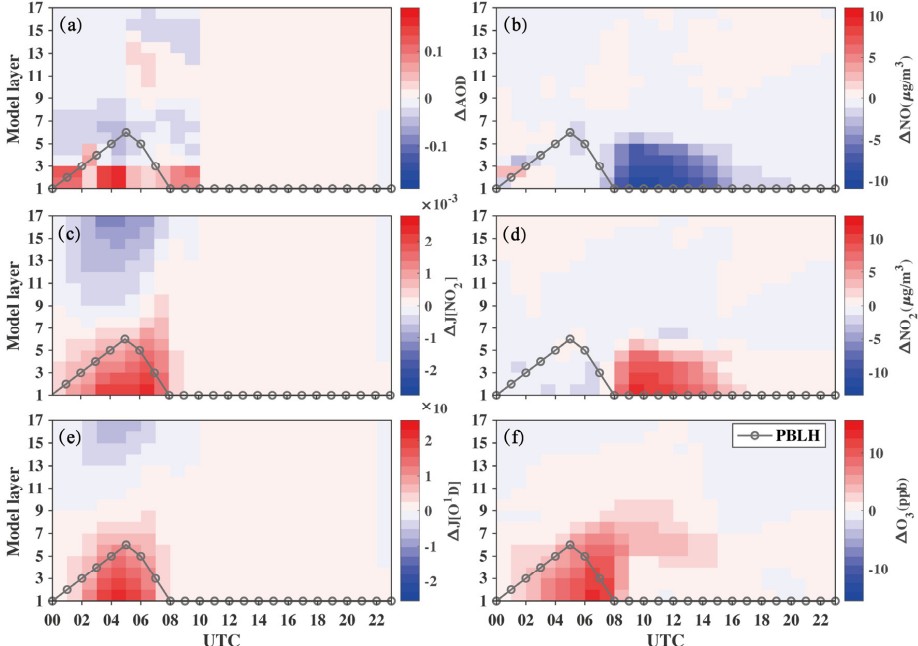

**Figure 7.** Same as Figure 6, but under conditions with lower planetary boundary layer heights (PBLH).

### 4.3 Impact of revised aerosol vertical distribution on PM$_{2.5}$

Modifying the vertical distribution of aerosols in the model resulted in significant changes in photolysis rates, which subsequently affected the atmospheric oxidizing capacity and, in turn, the chemical rate of secondary aerosol formation. In this section, we will discuss the impact of revised aerosol vertical distribution on atmospheric oxidizing capacity and PM$_{2.5}$ formation.

### 4.3.1 Wintertime

Figure 8 and Figure S6 display the spatial distribution of changes in surface PM$_{2.5}$ concentration and the relative changes, respectively. In winter, the increase in PM$_{2.5}$ concentration was primarily concentrated in the modified region, with high-value areas corresponding to the regions of NO$_2$ increase (Fig. 8a). PM$_{2.5}$ concentration was increased by an average of 0.8 µg m$^{-3}$ (0.2%), with a maximum increase of 3.1 µg m$^{-3}$ (1.2%) in the modified region (Table 1). The underestimation of





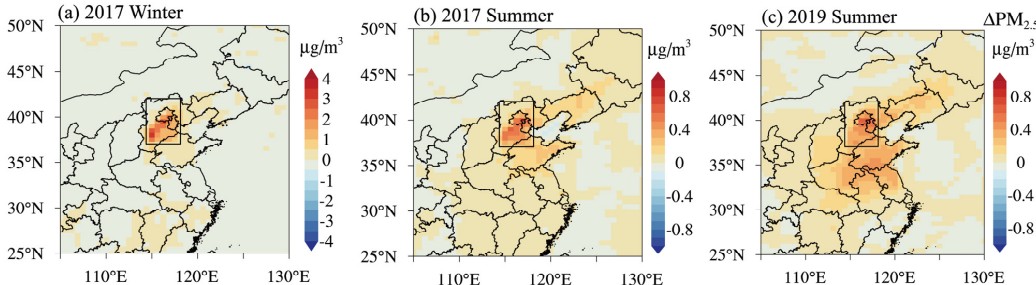


**Figure 8.** Simulated responses of PM$_{2.5}$ concentration to observational constraints on aerosol vertical distribution over North China. From left to right are for winter 2017 (a), summer 2017 (b), and summer 2019 (c), respectively. The black rectangle shows the region where vertical distribution modification was performed.

PM$_{2.5}$ in winter was slightly improved with observational constraints on vertical distribution (Table S1). The acceleration
of photolysis in the surface layer caused by modifications of aerosol vertical distribution enhanced the atmospheric oxidizing capacity and promoted secondary aerosol formation.

Figure 9 presents the changes in daily average PM$_{2.5}$ concentration and the relative changes along the longitude profile of the observation station. The increase in PM$_{2.5}$ in North China coincided with periods of enhanced photolysis and increased ozone in winter. As illustrated in Table 1, the regional average daily PM$_{2.5}$ changed by −0.1~2.9 µg m$^{-3}$ (−0.6~1.5%), and
the regional maximum daily PM$_{2.5}$ changed by 0.1–21.5 µg m$^{-3}$ (0.4–5.3%). The regional average API impact on PM$_{2.5}$ was −0.1~−4.4% in North China, which was reduced by approximately 30%. Although the impact of revised aerosol vertical distribution on seasonal average PM$_{2.5}$ was relatively small, its contribution during pollution events was notable. For example, on 25 January, the daily average PM$_{2.5}$ increased by 16.5 µg m$^{-3}$, 8.0 µg m$^{-3}$, 18.8 µg m$^{-3}$, and 10.3 µg m$^{-3}$ at Beijing, Tianjin, Baoding, and Langfang stations, respectively.



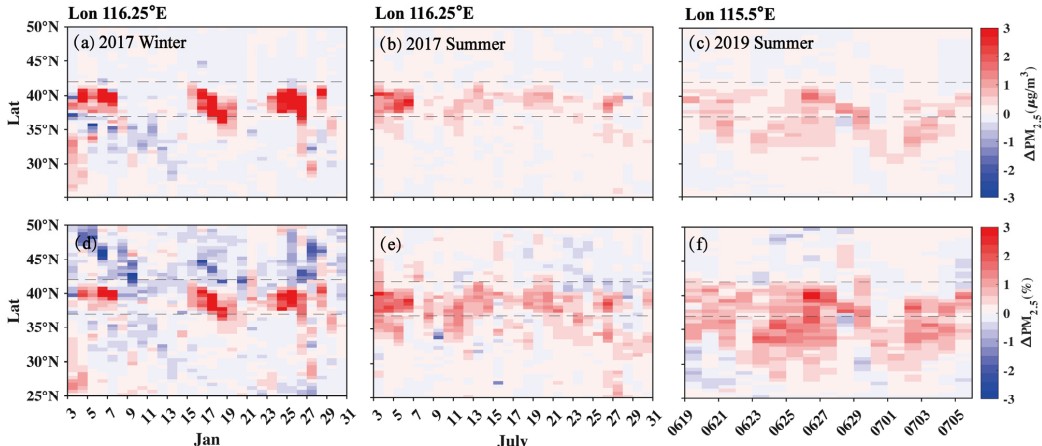


**Figure 9.** Simulated responses of daily average (daytime) PM$_{2.5}$ to observational constraints on aerosol vertical distribution along the longitude profiles of observation stations. Top panel: mass concentration (µg m$^{-3}$); Bottom panel: proportion (%). From left to right are for winter 2017 (a, d), summer 2017 (b, e), and summer 2019 (c, f), respectively. The dotted lines indicate the latitude range of the modified region, and the longitudes of the stations are marked at the top.

**4.3.2 Summertime**

In summer, the impact of aerosol vertical distribution on PM$_{2.5}$ concentration was a quarter of that in winter, with an average of 0.2 µg m$^{-3}$ and a maximum of 0.7 µg m$^{-3}$ in the modified region. However, the relative change in PM$_{2.5}$ was higher than in winter, with an average of 0.3%. The results from observational constraints based on radiosonde data also show an average of 0.2 µg m$^{-3}$ (0.5%) increase in PM$_{2.5}$ in summer 2019. Zhao et al. (2023) reported that the emission reduction in 385 the summer of 2019 can cause an increase of PM$_{2.5}$ by 0.54 µg m$^{-3}$ in the Beijing-Tianjin-Hebei region through API. This suggests that changes in aerosol vertical distribution are comparable to emission reduction when evaluating the effects of API on PM$_{2.5}$ in summer. Different from the wintertime, the differences in daily average changes of PM$_{2.5}$ between each day were relatively small (Fig. 9e–f). In summer 2017, the regional average changes in daily PM$_{2.5}$ and the regional maximum changes were 0–0.7 µg m$^{-3}$ (0–1.0%) and 0.2–2.9 µg m$^{-3}$ (0.5–2.8%). The increase in PM$_{2.5}$ over North China 390 in summer 2019 closely mirrored the values in 2017.



### 4.3.3 Role of atmospheric oxidizing capacity

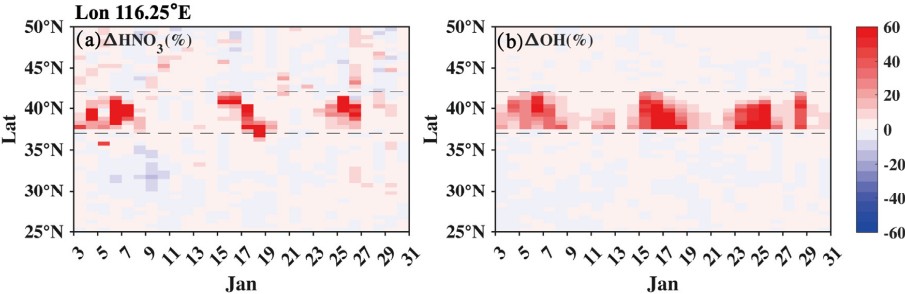

**Figure 10.** Simulated responses of daily average (a) HNO₃ and (b) OH to observational constraints on aerosol vertical
distribution along the longitude profiles of the lidar station in winter. The dotted lines indicate the latitude range of the
modified region, and the longitude of the station is marked at the top.

To explore the impact of revised aerosol vertical distribution on PM$_{2.5}$, we further analysed the major components
contributing to the PM$_{2.5}$ increase. Secondary aerosols can account for nearly all of the PM$_{2.5}$ increase, and their formation
is strongly influenced by the atmospheric oxidizing capacity. We compared the relative changes of hydroxyl radicals (OH)
in Fig. S7. It is found that OH increased by 13.5% in winter and 14.2% in summer over North China, indicating no
significant seasonal differences. As we have mentioned in Section 4.1, $J[O^1D]$ increased 73.8% in winter and 41.8% in
summer, which promoted the OH production. In winter, OH formation from the photolysis of HONO is also important
(Tan et al., 2018). The revised aerosol vertical distribution accelerated this process, resulting in a regional average reduction
of 4.4% in HONO concentration. Therefore, the atmospheric oxidizing capacity was enhanced by the increase in these OH
source reactions. However, the acceleration of $J[O^1D]$ and the photolysis of HONO were more significant in winter than
in summer, with a similar increase in OH concentration, suggesting that the oxidation of NO₂ by OH was promoted more
in winter.

Among secondary aerosols, nitrate was the major contributor to the increase of PM$_{2.5}$ in both winter and summer,
accounting for approximately 75%. The formation of nitrate during the daytime is primarily contributed by the aerosol
uptake of nitric acid (HNO₃) (Lu et al., 2019), which is mainly from the oxidization of NO₂ with OH. The enhancement of
photolysis led to an increase in OH during both summer and winter; however, with abundant NO$_x$ in the atmosphere during
winter, more NO$_x$ reacted with OH to form HNO₃. Figure 10 and Figure S8 show the changes in HNO₃ and OH in winter
and summer, respectively. Corresponding to the periods with OH increase, HNO₃ concentration increased significantly in
winter, reaching up to 60%. The areas with remarkable relative changes in HNO₃ concentration aligned with the increase





of $NO_2$ in Fig. 5a and the strong response of $PM_{2.5}$ in Fig. 9a. In summer, the intensity of atmospheric oxidizing capacity
was stronger than in winter (Liu et al., 2021b) and the $NO_x$ concentration was lower; as a result, $HNO_3$ increased less than
10% even though the increase in OH was similar to winter. In general, the enhancement of atmospheric oxidizing capacity,
combined with high $NO_x$ emissions, contributed to the strong response of $PM_{2.5}$ to observational constraints on aerosol
vertical distribution during winter pollution events.

## 5 Conclusions

In this study, we quantified the response of atmospheric photochemistry to observational constraints on aerosol vertical
distribution in different seasons using continuous observations from ground-based lidar and radiosonde. Sensitivity
experiments were conducted by modifying the aerosol vertical distribution over North China in the photochemical module
of the GEOS-Chem model, and the simulations in winter 2017 and summer 2017 were based on lidar data and the
simulations in summer 2019 were based on radiosonde data.

Compared to the observations, the GEOS-Chem model underestimated the AOD by 20.8%–30.1% below layer 2 (~0.19
km) in winter and by 3.4%–43.9% below layer 6 (~0.73 km) in summer, while overestimating AOD above these layers.
After observational constraints, the AOD increased in the lower layers and decreased in the upper layers. The changes of
photolysis rates of $NO_2$ and ozone ($J[NO_2]$ and $J[O^1D]$) were consistent with the vertical changes in AOD in both seasons.
The regional average $\Delta J[NO_2]$ and $\Delta J[O^1D]$ were 63.4% and 73.8% in winter, 33.4% and 41.8% in summer. In terms of
time series, the photolysis rates primarily increased during pollution events in winter, reaching a maximum increase of over
100%. In contrast, the daily average photolysis rate changes in summer were more steady, with lower peaks than in winter.

Surface ozone increased due to the enhancement of photolysis rates in the modified region, with an average of 0.9 ppb and
0.5 ppb in winter and summer, respectively, which is about half of the default impact of API on ozone. During pollution
episodes, the regional average daily ozone increased by 0.1–3.4 ppb in winter and 0–1.9 ppb in summer following the
enhancement of photolysis; the regional maximum daily changes of ozone reached 10.5 ppb and 7.1 ppb in winter and
summer, respectively. The compensatory effect of PBL mixing under conditions with high PBLH could weaken the impact
of revised aerosol vertical distribution on surface ozone. The results from observational constraints based on radiosonde
data in summer 2019 were consistent with those in summer 2017, consolidating the reliability of our work. Additionally,
the model underestimation of ozone in North China was also greatly reduced during winter after the modification of aerosol
vertical distribution.

Due to the promotion of secondary aerosol formation under more active photochemistry, regional $PM_{2.5}$ concentration
increased by 0.8 µg m$^{-3}$ (0.2%) in winter and 0.2 µg m$^{-3}$ (0.3%) in summer. The response of $PM_{2.5}$ was weaker in summer,



with an increase of the daily average by 0–0.7 µg m$^{-3}$, which was close to the results based on radiosonde constraints. However, in winter, the regional average daily PM$_{2.5}$ changed by −0.1–2.9 µg m$^{-3}$, with a maximum of 8.0–18.8 µg m$^{-3}$ at

some city stations, indicating the importance of aerosol vertical distribution during the pollution period. The increase in nitrate concentration resulting from enhanced HNO$_3$ formation was the primary reason of the increase in PM$_{2.5}$ during winter. The increase in atmospheric oxidizing capacity can account for the strong response of PM$_{2.5}$ during heavy pollution.

Early studies based on box models have emphasized that API can lead to great reductions in photolysis rates and ozone concentrations (Hollaway et al., 2019; Wang et al., 2020b). They found a greater reduction in the net photochemical

production of vertical ozone than surface ozone. Recent three-dimensional chemical transport model studies have highlighted the importance of the overlooked process in box model studies—boundary layer mixing, which can account for this inconsistency (Gao et al., 2020; Yan et al., 2023). Our study, based on observational constraints of aerosol vertical distribution, resolves an underappreciated issue and provides a deeper understanding of the impact of API. When considering observational constraints, the impact of API on surface ozone decreased significantly by 36%–56%, suggesting

that the impact of API in previous studies would be overstated. Despite the weakening of API, revising aerosol vertical distribution can inspire more accurate predictions of ozone levels and the atmospheric oxidizing capacity in winter, particularly in regions with complex aerosol vertical distributions. The comparison of cases with different boundary layer heights also offered valuable insights into the discrepancies in changes in photolysis rate and surface ozone.

In future studies, more efforts are needed to improve the model performance in aerosol vertical distribution. Although the

vertical distribution was constrained, discrepancies still exist between the simulated AOD and the observations. More accurate aerosol optical properties, such as single-scattering albedo (SSA), which can help distinguish between the scattering and absorption of light, are strongly needed in the model simulation. In summary, our results confirmed the importance of reasonable representativeness of aerosol vertical distribution in API effects and provided valuable implications for future studies.

**Data availability**

All data in this article are available on request from the first author (chenxi@nuist.edu.cn).

**Author contributions**

KL and XC designed the research. XC conducted the analysis with the help from TY, XJ, and LC. SZ was in charge of data curation. XC wrote the draft manuscript. KL, YY, BH, BZ, ZW, and HL reviewed and edited the manuscript.



**Competing interests**

The authors declare that they have no conflict of interest.

**Acknowledgments**

The author Xi Chen would like to thank the support from the Postdoctoral Fellowship Program of CPSF under Grant Number GZC20240733 and the Jiangsu Funding Program for Excellent Postdoctoral Talent. We would like to greatly thank Professors Xuhui Cai, Yu Song, and Hongsheng Zhang from Peking University for their efforts in collecting the radiosonde data.

**Financial support**

This research was supported by the National Natural Science Foundation of China (grants 42293323 and 42205114), the Natural Science Foundation of Jiangsu Province (BK20240035).

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
