# Peer review of "Simulated photochemical response to observational constraints on aerosol vertical distribution over North China"

_EGUsphere, 2025_

## Author Comment (AC1)

**Response to Referee #1**

**General Comments:**

This paper presents a modelling study that investigates the impact of the observed vertical aerosol distribution on the simulated photochemistry over Northern China. This paper builds on gaps in the field where few modelling studies using chemical transport models have considered actual observed vertical aerosol distributions when simulating the effects of the aerosol - photolysis interaction. Using combinations of lidar data and radiosonde observations the authors demonstrate that incorporating these into GEOS-Chem can help address the biases in AOD in the model and show the subsequent response on ozone levels and particulate matter. The authors test the sensitivities of aerosol photolysis interactions using numerous scenarios and robustly evaluate the effects of including the observational constraints including looking at impacts of on the atmospheric oxidising capacity and boundary layer effects. They also evaluate the effects in both the winter and summer seasons where the impacts of aerosols on atmospheric chemistry can behave very differently. Overall, the paper is well written and provides a valuable contribution to understanding photochemistry under changing pollutant regimes and the importance of representing the vertical distribution of aerosols in the model as accurately as possible. I recommend publication in ACP after the following minor comments are addressed.

We thank the referee for the constructive comments, and we have addressed them carefully. Please find our response in blue.

**Specific Comments:**

This paper focuses on the impacts of aerosol optical depth on photochemistry. Whereas the authors provide evaluation of the model for aerosol optical depth (and identify and correct biases) they do not evaluate and benchmark how well it currently captures observed photolysis rates at the observational sites. I appreciate that measured photolysis rates (E.g. JNO2) may not be available at all sites but if there is data available it would be great if the authors could provide an observational comparison. Due to the potential of data limitations I would not suggest this is needed for publication but think it would greatly enhance the model evaluation if such data was available.

Many thanks for this suggestion, and we agree with the reviewer that a model evaluation against measured photolysis rates will be very helpful.

However, such a comparison during the study period is not available due to data limitations, although we have compared the simulated $J[NO_2]$ with the reconstructed dataset derived from observed data in Figure 3. This dataset was reconstructed by observational $J[NO_2]$, total ultraviolet radiation, and troposphere ultraviolet and visible (TUV) radiation model, and showed a great consistency with observational $J[NO_2]$ at Xianghe station ($R^2 = 0.94$). As demonstrated in Lines 274–276, the revised aerosol vertical distribution improved the underestimation of $J[NO_2]$.

Besides, following Referee #2's suggestion, we have added additional $NO_2$ validation in Lines 155–161, to further verify the reliability of the $NO_2$ photolysis simulation.

We also added a discussion to highlight the necessity of including more model evaluations using observed photolysis rates in future studies in Lines 486–488: "Moreover, the model validation with observed photolysis rates should be involved, as the photolysis rate is one of the most direct variables for verifying the simulation of photochemical reactions."

**Technical Corrections:**
Page 3, line 64: Please remove 'the' between suppress and secondary
Corrected.

Page 3, line 94: This last sentence is not quite clear. Do you mean to investigate the response of photochemistry to observed constraints on vertical aerosol distributions?
Yes. It has been changed to: "to investigate the response of photochemistry to observed constraints on aerosol vertical distributions."

Page 4, line 122: You mention the various schemes used here for boundary layer mixing, dry deposition, wet deposition and emissions but you don't mention the chemical scheme deployed in the model. Please add this here.
We have added the description of the chemical scheme in Lines 124–125: "The standard chemistry simulation with fully coupled ozone-$NO_x$-hydrocarbon-aerosol chemistry mechanisms was employed in this study."

Page 5, Figure 1: I am finding the red boxes in plots (b) and (c) a little hard to read, would it be possible for these to be made a little bolder and perhaps plotted in black to make them stand out better on the page?
Thanks! We have changed the boxes in Figure 1(b) and (c) using bolder black ones.

---

## Author Comment (AC2)

**Response to Referee #2**

The manuscript titled "Simulated photochemical response to observational constraints on aerosol vertical distribution over North China" investigate the response of photochemistry to observational constraints on aerosol vertical distribution by applying lidar and radiosonde observations and the chemical transport model (GEOS-Chem). This study provides better understanding on the impact of aerosols on $O_3$ pollution. The article is interesting. It can be accepted after fully considering the following suggestions.

We thank the referee for the very helpful comments and suggestions. We have addressed them carefully and please find our response in blue.

The aerosol vertical distribution may differ among different sites such as urban areas and rural areas. This depends on meteorological factors and local emissions levels. Here the authors use the measurements at two sites to represent the whole North China region, which may lead to bias. According to Figure 2, it seems that the revised aerosol vertical distribution for the region is significant different from the measurements. More evidence should be given for the spatial representativeness of the measurements.

We agree with the reviewer's comment. The comparison of $PM_{2.5}$ profiles observed simultaneously at six radiosonde stations was added to the manuscript to confirm the spatial representativeness of the observations.

Lines 214–217: "Figure S5 shows the vertical distribution of $PM_{2.5}$ concentrations in model layers at six radiosonde stations for winter 2019. According to Li et al. (2022), these stations can be spaced up to 150 km apart, and simultaneous observations for a consecutive month show that the timing of occurrence and vertical profiles of $PM_{2.5}$ concentrations at different stations are very similar."

Lines 217–221: "The differences in real aerosol vertical distribution between the observation station and the modified region are inevitable. However, the consistency of pollutant concentrations at surface observation stations (Table S2), the regional characteristic of vertical distributions from satellite observations in Figure 1, the similarity of simulated aerosol vertical distributions in the modified region, combined with the $PM_{2.5}$ profiles at six radiosonde stations, could demonstrate the spatial representativeness of the measurements."

The difference between the revised aerosol vertical distribution and the observation contributed most to the bias of total AOD. We have discussed the influence of the accuracy of the AOD simulation on the revised aerosol vertical distribution in Lines 256–262 (Figure S7).

[Figure]

**Figure S5**: (a) Spatial distribution of radiosonde stations. BD: Baoding; BZ: Bazhou; CZ: Cangzhou; DZ: Dingzhou; RQ: Renqiu; TN: Tuonan. (b–g) Vertical distribution of PM2.5 concentrations in model layers from 26 November 2019 to 26 December 2019 at six radiosonde stations.

The methodology of the radiative transfer module you used should be depicted in detail. Besides AOD, single scattering albedo (SSA) is another important aerosol optical property that affects photolysis rates. I suggest the authors to provide validation and discussion of SSA in the manuscript.

More details about the radiative transfer module have been added in Lines 127–129:

"In GEOS-Chem, the photolysis rates are calculated by the updated Fast-JX v7.0 scheme (Eastham et al., 2014), which accounts for the presence of aerosols, clouds, and absorbing gases. Developing from the Fast-J radiative transfer algorithm (Wild et al., 2000), FAST-JX provides a full scattering calculation for 18 wavelength bins covering 177–850 nm. The photolysis rates are determined by the resolved flux and cross sections of different species in each wavelength bin. The impacts of aerosols, including sulfate–nitrate–ammonium (SNA), organic aerosol (OA), black carbon (BC), sea salt, and dust, on photolysis rates are closely related to aerosol optical properties, which are calculated using the Mie code based on the simulated three-dimensional aerosol concentrations."

We have also validated SSA at 440 nm with Aerosol Robotic Network (AERONET) data in Lines 169–171: "As photolysis rates can also be affected by the single scattering albedo (SSA), we validated the simulation of SSA at 440 nm with the observation from AERONET in January 2017 (Figure S1). It shows that SSA was well reproduced, with an NMB of 3.8%, so only AOD profiles were revised in this work."

[Figure]

**Figure S1**: Observed single scattering albedo (SSA) from AERONET and simulated SSA from GEOS-Chem at 440 nm for winter 2017. The observation station is located in Beijing (39.977°N, 116.381°E).

Wild, O., Zhu, X., and Prather, M. J.: Fast-J: Accurate Simulation of In- and Below-Cloud Photolysis in Tropospheric Chemical Models, Journal of Atmospheric Chemistry, 37, 245-282, https://doi.org/10.1023/A:1006415919030, 2000.

For the discussion of atmospheric oxidizing capacity, HONO is an important OH radical source. Nevertheless, HONO has multiple sources and most air quality models can not fully consider these sources and thus underestimate HONO concentrations. I wonder which sources of HONO are considered in the GEOS-Chem model, and the resulting potential uncertainty should be clarified.

We have clarified the HONO sources in the GEOS-Chem model and pointed out the potential uncertainty caused by HONO underestimation in Lines 423–427: "Herein, we added the emission of HONO from transportation by applying a HONO/$NO_x$ emission ratio (Zhang et al., 2019a), and the heterogeneous reaction of $NO_2$ (Shah et al., 2020) was also included in the GEOS-Chem model, but the other HONO sources like soil emissions (Tan et al., 2023), livestock farming (Zhang et al., 2023), and particulate nitrate photolysis (Andersen et al., 2023) were not included. It should be noted that the enhancement of atmospheric oxidizing capacity may be weakened due to the underestimation of HONO concentrations in the model."

References:
Andersen, S. T., Carpenter, L. J., Reed, C., Lee, J. D., Chance, R., Sherwen, T., Vaughan, A. R., Stewart, J., Edwards, P. M., Bloss, W. J., Sommariva, R., Crilley, L. R., Nott, G. J., Neves, L., Read, K., Heard, D. E., Seakins, P. W., Whalley, L. K., Boustead, G. A., Fleming, L. T., Stone, D., and Fomba, K. W.: Extensive field evidence for

the release of HONO from the photolysis of nitrate aerosols, Science Advances, 9, https://doi.org/10.1126/sciadv.add6266, 2023.

Shah, V., Jacob, D. J., Li, K., Silvern, R. F., Zhai, S., Liu, M., Lin, J., and Zhang, Q.: Effect of changing NOx lifetime on the seasonality and long-term trends of satellite-observed tropospheric NO2 columns over China, Atmospheric Chemistry and Physics, 20, 1483-1495, https://doi.org/10.5194/acp-20-1483-2020, 2020.

Tan, W. S., Wang, H. L., Su, J. Y., Sun, R. Z., He, C., Lu, X., Lin, J. T., Xue, C. Y., Wang, H. C., Liu, Y. M., Liu, L., Zhang, L., Wu, D. M., Mu, Y. J., and Fan, S. J.: Soil Emissions of Reactive Nitrogen Accelerate Summertime Surface Ozone Increases in the North China Plain, Environmental Science & Technology, https://doi.org/10.1021/acs.est.3c01823, 2023.

Zhang, J. W., An, J. L., Qu, Y., Liu, X. G., and Chen, Y.: Impacts of potential HONO sources on the concentrations of oxidants and secondary organic aerosols in the Beijing-Tianjin-Hebei region of China, Science of The Total Environment, 647, 836-852, https://doi.org/10.1016/j.scitotenv.2018.08.030, 2019.

Zhang, Q., Liu, P. F., Wang, Y., George, C., Chen, T. S., Ma, S. Y., Ren, Y. A., Mu, Y. J., Song, M., Herrmann, H., Mellouki, A., Chen, J. M., Yue, Y., Zhao, X. X., Wang, S. G., and Zeng, Y.: Unveiling the underestimated direct emissions of nitrous acid (HONO), Proceedings of the National Academy of Sciences, 120, https://doi.org/10.1073/pnas.2302048120, 2023.

For the change in PM2.5, the authors have discussed the change of nitrate and HNO3. How about other secondary aerosol components such as sulfate and SOA?

We have summarized the contributions of other secondary aerosol components in Lines 431–432: "Next come ammonium and sulfate, contributing approximately 20% and 8% respectively, while secondary organic aerosols (SOA) can be ignored."

Line 150, Line 195: For the correlation analysis, the time and spatial resolution of data point and the number of data should be depicted. This should be double checked in anywhere else in the manuscript.

Thanks. We have reviewed the manuscript, and the details of the data have been well depicted.

Lines 154–155: "These statistics were calculated based on the average hourly concentrations at 12 observation stations in the region (37–42°N, 114.375–118.125°E) where modification of aerosol vertical distribution was applied."

Lines 203–205: "Based on the observed surface pollutant concentrations, we found a strong temporal consistency between hourly concentrations at vertical observation stations and the average hourly concentrations at 12 city stations in the modified region (Table S2)."

Lines 210–212: "Besides, we analysed the correlation of simulated daily average aerosol vertical distributions at the observed grid point and the other 76 grid points in the modified region, finding that over half of the grid points exhibited significant correlation (Fig. S4)."

The authors have validated O3 and PM2.5 by comparison with measurements. I suggest to additionally validate NO2 given that NO2 is a key precursor of O3 and PM2.5.

Thank you for the suggestion. We have added the validation of $NO_2$ in the manuscript and Table S1.

Lines 155–158: "The GEOS-Chem generally reproduced the variation in surface concentrations in winter, achieving correlation coefficients of 0.75, 0.69, and 0.69 for $NO_2$, ozone, and $PM_{2.5}$, with underestimations of 22.1%, 24.8%, and 4.3% for $NO_2$, ozone, and $PM_{2.5}$, respectively."

Lines 160–161: "The model simulated $NO_2$ and ozone best in the summer of 2019, with an NMB of −2.8% and −0.9%, but there was an overestimation of $PM_{2.5}$."

**Table S1:** Model performance metrics of BASE and REGION simulations for $NO_2$, ozone, and $PM_{2.5}$ in North China. All statistics were calculated from the average of observation stations in the modified region.

| | | 2017 Winter | | 2017 Summer | | 2019 Summer | |
|---|---|---|---|---|---|---|---|
| | | BASE | REGION | BASE | REGION | BASE | REGION |
| $NO_2$ | R | 0.75 | 0.76 | 0.49 | 0.49 | 0.61 | 0.60 |
| | NMB (%) | -22.1 | -21.6 | -28.3 | -29.8 | -2.8 | -4.8 |
| Ozone | R | 0.69 | 0.65 | 0.81 | 0.80 | 0.84 | 0.83 |
| | NMB (%) | −24.8 | −16.6 | 44.2 | 45.9 | −0.9 | 0.4 |
| $PM_{2.5}$ | R | 0.69 | 0.69 | 0.35 | 0.35 | 0.80 | 0.80 |
| | NMB (%) | −4.3 | −3.7 | 21.3 | 21.9 | 49.2 | 49.9 |